# Emerging Perspectives on the Antiparasitic Mebendazole as a Repurposed Drug for the Treatment of Brain Cancers

**DOI:** 10.3390/ijms24021334

**Published:** 2023-01-10

**Authors:** Daniela Meco, Giorgio Attinà, Stefano Mastrangelo, Pierluigi Navarra, Antonio Ruggiero

**Affiliations:** 1Pediatric Oncology Unit, Fondazione Policlinico Universitario A. Gemelli IRCCS, 00168 Rome, Italy; 2Dipartimento Scienze della Salute della Donna, del Bambino e di Sanità Pubblica, Università Cattolica del Sacro Cuore, 00168 Rome, Italy; 3Department of Healthcare Surveillance and Bioethics, Section of Pharmacology, Università Cattolica del Sacro Cuore-Fondazione Policlinico Universitario A. Gemelli IRCCS, 00168 Rome, Italy

**Keywords:** Mebendazole, drug repurposing, antitumor therapy, brain cancer, benzimidazole

## Abstract

Repurposing approved non-antitumor drugs is a promising and affordable strategy in drug discovery to identify new therapeutic uses different from the original medical indication that may help increase the number of possible, effective anticancer drugs. The use of drugs in ways other than their original FDA-approved indications could offer novel avenues such as bypassing the chemoresistance and recurrence seen with conventional therapy and treatment; moreover, it can offer a safe and economic strategy for combination therapy. Recent works have demonstrated the anticancer properties of the FDA-approved drug Mebendazole. This synthetic benzimidazole proved effective against a broad spectrum of intestinal Helminthiasis. Mebendazole can penetrate the blood–brain barrier and has been shown to inhibit the malignant progression of glioma by targeting signaling pathways related to cell proliferation, apoptosis, or invasion/migration, or by increasing the sensitivity of glioma cells to conventional chemotherapy or radiotherapy. Moreover, several preclinical models and ongoing clinical trials explore the efficacy of Mebendazole in multiple cancers, including acute myeloid leukemia, brain cancer, oropharyngeal squamous cell carcinoma, breast cancer, gastrointestinal cancer, lung carcinoma, adrenocortical carcinoma, prostate cancer, and head and neck cancer. The present review summarizes central literature regarding the anticancer effects of MBZ in cancer cell lines, animal tumor models, and clinical trials to suggest possible strategies for safe and economical combinations of anticancer therapies in brain cancer. Mebendazole might be an excellent candidate for the treatment of brain tumors because of its efficacy both when used as monotherapy and in combination as an enhancement to standard chemotherapeutics and radiotherapy, due to its effectiveness on tumor angiogenesis inhibition, cell cycle arrest, apoptosis induction, and targeting of critical pathways involved in cancer such as Hedgehog signaling. Therefore, attention to MBZ repurposing has recently increased because of its potential therapeutic versatility and significant clinical implications, such as reducing medical care costs and optimizing existing therapies. Using new treatments is essential, particularly when current therapeutics for patients with brain cancer fail.

## 1. Introduction

Cancer ranks as a leading cause of death and a significant barrier to increasing life expectancy in every country [1]. According to estimates from the World Health Organization (WHO) in 2019, cancer is the first or second leading cause of death before age 70 in 112 of 183 countries and ranks third or fourth in 23 countries [2]. Current tumor treatments usually consist of chemotherapy, radiotherapy, surgery, targeted therapy, immunotherapy, and endocrine therapy that can be used individually or in combination, depending on the stage and type of tumor and diagnosis. Several factors reduce the practical improvement of a cancer patient’s prognosis. Drug-untreatable targets, chemoresistance, tumor heterogeneity, and metastases formation are significant barriers to the effective cure of cancer patients. Despite intense preclinical and clinical research efforts, the survival rate of patients suffering from the most aggressive cancer types has not yet improved, mainly due to therapeutic failure. Moreover, nearly all anticancer medications currently on the market have serious adverse effects; therefore, new, safer anticancer drugs are desirable.

However, developing new therapeutics has become increasingly difficult for pharmaceutical companies; indeed, the discovery and development of new drugs represent long and expensive processes with a low overall probability of success. At present, it takes about 10–15 years to develop a new drug [3], and despite all the effort, the success rate remains very low (around 2%) [4,5]. In recent years, the pool of newly discovered and Food and Drug Administration (FDA)-approved drugs has diminished [6], primarily due to the increasing cost of clinical trials [7]. From 2004 through 2012, the median price for a pharmaceutical clinical trial is around USD 2–3 billion. In this scenario, new approaches to drug use can reduce development time and offer novel and effective anticancer therapies. Compounds clinically used for non-cancer indications could lead to safe, affordable, and timely treatment options for patients with high medical needs.

In recent years, the concept of drug repurposing (also known as “drug repositioning”, “reprofiling”, or “retasking”) is gaining increasing attention as a strategy for identifying new therapeutic applications of existing drugs or those under investigation outside the scope of the original medical indication [8,9,10,11]. Many researchers have demonstrated that non-traditional antitumor drugs have distinct anticancer actions and represent a safe and promising treatment option. This strategy also provides an opportunity to develop innovative formulations, which may provide clinical benefits compared to older marketed drugs [12]. The benefits of this strategy are that the risk of failure is lower than the approach with newly developed drugs. The repurposed drug is sufficiently safe in preclinical models and humans. If early-stage trials are complete, it is less likely to fail, at least from a safety point of view, in subsequent efficacy trials. Another benefit of this strategy is that it can reduce the time frame for drug development because most preclinical testing, safety assessment, and in some cases, formulation development are already complete.

Consequently, less investment is needed, although this will vary greatly depending on the stage and development of the repurposing candidate [13,14]. The pharmacokinetic, pharmacodynamic, and toxicity profiles of approved medicines are well known, so the new users can more easily be translated into phase II and III clinical trials [15,16,17,18]. Moreover, drugs that have been on the market for several years are often relatively cheap compared to new medicinal products; furthermore, these medicines’ wide availability and affordability could facilitate clinical research and enable timely and affordable access for patients [19,20,21]. Finally, repurposed drugs may reveal new targets and pathways, suggesting further drug exploitation.

For this reason, there has been substantial interest in uncovering previously uncharacterized therapeutic properties among medications that the FDA has already approved. In this context, a relevant example of effective drug repurposing in cancer and advanced clinical trials is Mebendazole (MBZ). MBZ belongs to the anti-helminthic drugs category, including various chemical entities (like albendazole, parbendazole, fenbendazole, triclabendazole, oxibendazole, ricobendazole, and flubendazole) and has several distinct modes of action [22]. For decades, MBZ safely treated infections caused by parasitic worms (helminths) that colonize the human intestine, developed for treating veterinary parasites first and then for clinical application. The main reason for MBZ repurposing lies in its ability to inhibit both microtubule formation and glucose uptake; these aspects predetermine these compounds to possible cytostatic effects. In addition, several research groups investigated the repurposing of MBZ as a possible anticancer agent in oncology for its ability to interfere with several key oncogenic signal transduction pathways [23,24].

In this review, we highlight the potential anticancer effect of MBZ and its potential mechanistic actions on brain cancer. Several recent studies reported the anticancer effect of MBZ, with a wide range of pharmacological mechanisms against brain cancer, including proliferation inhibition, cell cycle arrest, apoptosis induction, angiogenesis inhibition, and targeting of several key oncogenic signal transduction pathways. In addition, we summarize the pharmacokinetic properties and side effects of MBZ, aiming to clarify the basic understanding and improve its use and potential in brain cancer therapy. Possible advantages, limitations, and additional perspectives are discussed and evaluated.

## 2. Physiopathology of Glioma Development

Diffuse gliomas represent 80% of malignant brain tumors [25]. The standard classification of brain tumors by the World Health Organization (WHO, Geneva, Switzerland) is based on their mitotic activity, proliferation, and degree of necrosis; they are identified in four grades:Grade I gliomas are low-grade benign lesions, such as pilocytic astrocytomas, that have limited proliferation and are frequent in children; however, they might acquire higher levels of malignancy. Surgical resection is a good therapeutic option in most cases.Grade II gliomas have low-grade but infiltrative lesions that tend to show higher recurrence after surgical resection.Grade III gliomas have intermediate- to high-grade lesions with atypia, higher mitotic activity, and evidence of malignancy. Patients with these tumors usually are prescribed additional radiation as well as chemotherapy.Grade IV tumors are high-grade malignant lesions with higher mitotic activity, microvascular proliferation, high necrosis, and the worst prognosis.

In 2016, CNS WHO released the latest update of tumor classification on the basis of the latest studies on molecular characterization [26]. Genomic sequencing efforts provided unprecedented insight into the genomic aberrations and cellular signaling mechanisms that drive these cancers, and discoveries from these efforts translated into novel diagnostic algorithms, biomarkers, and therapeutic strategies in neuro-oncology. These sequencing efforts helped revise WHO classification of nervous system tumors, and histologic and molecular parameters were incorporated into the diagnostic catalog [26,27]. Several diagnoses were restructured (diffuse gliomas, medulloblastomas, and embryonal tumors), and novel tumor entities were introduced. For this purpose, researchers at WHO used the datasets generated by The Cancer Genome Atlas (TCGA) and related gene expression and DNA methylation signatures with prognosis. Compared to the previous release, the principal change in the new classification is the relevant reorganization of diffuse gliomas, according to the IDH status, co-deletion of chromosome arms 1p and 19q, and MGMT methylation. The status of isocitrate dehydrogenase 1 and 2 (IDH1/2) enzymes are key reference genes for brain tumor classification. Mutations of IDH1 and IDH2 lead to a favorable outcome [28] and have more prolonged overall survival in comparison to those with IDH wild type. High-grade gliomas with IDH wild type have shorter overall survival, while low-grade gliomas display a poor glioblastoma (GBM)-like trajectory. The new diagnostic criterion for oligodendroglioma is the allelic loss of chromosomes 1p and 19q [29]. Although MGMT is not considered a valid predictive biomarker in diffuse gliomas, it is a prognostic biomarker (Figure 1) [30].

As for signaling pathways, brain tumors manifest aberrations in key cellular pathways that mediate cell growth and replication proliferation, survival, migration, and apoptosis, as observed in other cancers. These pathways are not linear sequences of enzymes but rather complex signaling pathways influenced by multiple growth factors. The Cancer Genome Atlas classified three main signaling pathways in gliomas’ pathogenesis: RTK/RAS/PI3K (receptor tyrosine kinase, RAS, phosphatidylinositol 3-kinase), p53, and retinoblastoma (RB); later, angiogenesis was added as an additional signaling pathway [26,31,32].

The PI3K/AKT/mTOR signal transduction pathway and the Ras/MAPK pathway are hyperactivated in several cancer types, including glioblastoma, and play a key role in tumorigenesis by activating the tumor promoters and inhibiting the tumor suppressors. Several growth factors activate PI3K (e.g., human EGFR family and platelet-derived growth factor receptor (PDGFR) family growth factors). RTKs are frequently activated in malignant gliomas, such as EGFR gene amplification, which occurs in approximately 40% of patients with GBM, and PDGFRA gene amplification, which occurs in up to 16% of GBM [33,34,35,36]. At the same time, the EGFR variant (EGFRvIII) shows ligand-independent constitutive activation of the receptor. This deletion mutant is observed in approximately 30% to 50% of EGFR-amplified gliomas [37]. Downstream signaling from growth factor receptors can be activated by loss or mutation in the neurofibromatosis 1 gene (NF1), mutations in KRAS, mutations in PIK3CA (the gene for PI3K), and deletion or loss of heterozygosity of PTEN. PI3K participates in the phosphorylation of AKT and promotes cellular proliferation by inactivating cell cycle inhibitors and promoting cell cycle proteins. The PTEN tumor suppressor gene is inactivated in 40–50% exclusively in primary glioblastomas and usually inhibits the PI3K/Akt pathway [38].

The Tumor Protein p53 (TP53) tumor suppressor gene encodes a protein that regulates several cellular processes, including cell cycle arrest, the response of cells to DNA damage, senescence, apoptosis, and cell differentiation and neovascularization. The p53 cascade regulates over 2500 genes, 67 of which are involved in tumorigenesis [39,40,41,42]. After DNA damage, TP53 is activated and induces transcription factors to regulate the expression of downstream effector genes to determine cell fate. The p53 pathway is the most altered in sporadic glioma, with some aberration of p53 signaling found in 87% of tumors [38]. These aberrations are most prevalent in low-grade glioma of astrocytic lineage and secondary glioblastoma, often concurrent with IDH mutation [43]. Generally, P53 inactivation in glioblastoma occurs through amplification of MDM2 (mouse double-minute 2) (11%) or MDM4 (4%), deletion of ARF (55%), and mutations of p53 itself [4]. MDM2 is a primary negative regulator of the p53 pathway; it provides transcriptional inhibition by directly binding p53 and degrades p53 through its ligase activity [44].

RB is a tumor suppressor gene encoding the retinoblastoma susceptibility protein 1 (RB1). The Rb pathway is an important cell cycle regulator that inhibits the entry of cells through G1 into the S-phase of the cell cycle. The Rb pathway is implicated in the malignant progression of astrocytoma and is commonly altered in glioblastoma (79%). When phosphorylated by cyclin D, cyclin-dependent kinase 4 (CDK4), and CDK6, RB1 will be inactive, resulting in an unregulated progression through the cell cycle and tumor growth [45] (Figure 2).

## 3. Mebendazole: Chemical Structure and Pharmacokinetic Properties

Mebendazole (methyl 5-benzoyl-1H-benzimidazol-2-yl-carbamate) appeared in 1968 as an agent active against a wide range of broad-spectrum anthelmintics applied first to human subjects in 1971 [46]. Table 1 and Table 2 summarize the pharmacokinetic properties of MBZ. MBZ acts directly on luminal parasites in the gastrointestinal tract. First-pass metabolism of MBZ ensures that only about 20% of the oral dose reaches systemic circulation, with maximum plasma concentration reached 2–4 h post-administration. After the first pass in the intestinal wall and liver, the metabolites are active against parasites in internal organs and tissues [47,48]. MBZ is also extensively metabolized by the liver to amino and hydroxylated amino forms of the parent compound; nevertheless, the enzymes responsible have not been identified [49,50,51]. However, also P450 enzymes are involved in the process due to the documented inhibition with cimetidine [52]. The MBZ and its metabolites are excreted primarily in feces and about 2% in the urine [53]. MBZ induces the block of microtubule functions of parasites and mammalian cells through inhibition of polymerization of ß-tubulin into microtubules. The result is a loss of transport of secretory vesicles, a decrease in glucose uptake, and an increase in the use of stored glycogen [51,54].

The World Health Organization (WHO)-recommended dose of MBZ is 40–50 mg/kg/day for at least 3–6 months and two years for cystic and alveolar echinococcosis, respectively [60]. The dosage varies according to the type of helminthic infection to treat. Pinworms are treated with a single 100 mg treatment, whereas roundworms or hookworms are treated with 100 mg twice daily for three days. In the case of long-term administration for treating human cystic and alveolar echinococcosis (also known as hydatid disease), the treatment is generally well tolerated. Still, the specific treatment for some patients must be discontinued. For example, in one open-labeled observational study, the patients treated with MBZ for alveolar echinococcosis (average: 24 months) experienced few adverse reactions, and in only three patients (of 17), the treatment was changed to albendazole due to intolerable side effects (reversible alopecia, psychological disturbance, and drop in performance) [61]. Side effects associated with low- to high-dose regimens include abdominal discomfort, flatulence, diarrhea, neutropenia, marrow aplasia, alopecia, allergic reaction, and elevations in transaminase levels [60]. However, drug withdrawal spontaneously reverses these relatively rare and mild adverse effects [23,62]. One of the most significant adverse effects is hepatotoxicity, although its incidence is infrequent; Tolomei et al. [63] have reported a case of severe hepatitis after 18 days of MBZ administration in a patient with Gilbert’s syndrome affected by pinworm infestation. After a standard dose of MDZ (2 × 100 mg/d for three days every week) and after short-term treatment, acute hepatitis manifested, with a serum level of ALT (alanine aminotransferase) higher than that observed for higher doses. Hepatic enzymes returned to normal over the following five weeks. Moreover, a case series report supports its safety in breastfeeding, where no evidence of MBZ-related toxicity in infants of lactating mothers has been observed [64]. MBZ is a teratogen in experimental rats if given in very high doses, although not in rabbits [31]. In human patients, MBZ is contraindicated during pregnancy [65]. However, long-term MBZ treatment (50 mg/kg daily for 9–18 months) was safe for children with hydatid disease [66]. The combination of MBZ (>500 mg) and metronidazole (>500 mg) is prohibited because severe and rare fatal adverse events such as Stevens–Johnson syndrome (or toxic epidermal necrosis) may occur. The risk increased with increasing doses of metronidazole but not MBZ, and there may be a synergistic interaction between MBZ and metronidazole [67].

Another well-known property of MBZ is its relative insolubility in water and most organic solvents. The absorption rate of oral MBZ in the human intestine is about 1–5% [51]. The low water solubility can cause a reduction in drug absorption that, combined with the rapid drug degradation, could result in an inadequate in vivo drug concentration and failure to achieve the therapeutic dose. Moreover, oscillation in the drug plasma levels results in unforeseeable levels of bioavailability. All these effects cause a reduction in MBZ’s potency in cancer therapy; thus, modifications enhancing its availability will determine its potential use in oncology. Several researchers suggested strategies to improve the poor aqueous MBZ and anticancer effects [59].

Alavi and Shahmabadi provide an overview of the strategies adopted to improve low bioavailability. To overcome these limitations, the authors proposed using nanocarrier-based formulations, pro-drug formulations, and solid dispersions (SDs) to improve drug solubility [68]. To overcome poor aqueous solubility, de la Torre-Iglesias et al. [69] have prepared re-dispersible microparticles (RDM) containing MBZ for improved oral bioavailability and therapeutic activity using low doses of MBZ (5 mg/kg). These formulations could optimize anthelmintic efficacy and enhance MBZ concentrations in muscle and cysts. This aspect is crucial for inoperable or disseminated cases of hydatidosis or neuro-cysticercosis.

Furthermore, low doses of MBZ lead to low toxicity in humans. Münst et al. [70] have demonstrated that bioavailability may significantly improve by coadministration with a high-fat meal. Plasma concentrations remained at 17 nmol/L in three fasting volunteers treated with 1.5 g doses. Instead, when the treated volunteers had a standard breakfast, the plasma concentration rose to 91,112 and 142 nmol/L within 2 to 4 h. In another clinical study, Braithwaite et al. [55] showed that MBZ plasma concentration-time profiles differed considerably between patients after oral administration of MBZ. The authors monitored the plasma concentrations of MBZ and its metabolites in twelve patients after receiving 10 mg/kg, in subjects receiving their first dose of MBZ, and in subjects with chronic treatment. The elimination half-lives ranged from 2.8–9.0 h; the peak plasma concentrations ranged from 17.7 to 116.2 ng/mL for subjects receiving their initial dose and from 99.4 to 500.2 ng/mg for subjects on chronic treatment. Studies reported by Alavi et al. [68] showed that the plasma AUCTs for the main metabolites of MBZ (methyl 5-(α-hydroxybenzyl)-2-benzimidazole carbamate and 2-amino-5 benzoylbenzimidazole) were about five times larger than the plasma MBZ AUCT found in patients on chronic therapy, confirming that the gastrointestinal tract poorly absorbs MBZ.

In patients on chronic MBZ therapy, MBZ concentrations found in both liver tissue and parasite tissue were significantly higher than in adipose tissue. While the MBZ concentration in liver was higher, that in other tissues was lower than the plasma Cmax detected for each patient. To reduce the wide inter- and intra-individual variation in systemic availability of MBZ, the drug should be taken with food. For intravenous administration, no guidelines can be provided because there are no data on the pharmacokinetics of MBZ after intravenous infusion. Bekhti et al. [57] and Luder et al. [58] proved that MBZ produces pharmacodynamic effects possibly exaggerated by pharmacokinetic interactions with other agents. In that paper, they showed that cimetidine increases plasma levels of the MBZ (maximum serum levels rose from 55.7 ± 30.2 ng/mL [0.19 ± 0.10 μM] to 82.3 ± 41.8 ng/mL [0.28 ± 0.14 μM], on 1.5 g of MBZ following chronic dosing of cimetidine at 400 mg three times a day for 30 days). The available evidence suggests that the increase in Cmax of MBZ, caused by concurrent use of cimetidine, would influence the therapeutic actions of MBZ, probably due to inhibition of hepatic first-pass cytochrome P450-mediated metabolism of the MBZ. However, what remains unclear is whether such an increase in Cmax would predispose a patient to adverse effects. Due to such an interaction, clinicians should closely monitor patients receiving this combination for signs of adverse effects of MBZ and choose alternative agents when possible. In addition, Corti et al. [56] investigated the effect of ritonavir (protease inhibitor of the CYP3A system) on the pharmacokinetics of MBZ under single-dose and steady-state conditions. They observed that the pharmacokinetic parameters of MBZ did not change in the short-term administration of ritonavir. In contrast, long-term administration of ritonavir resulted in significant changes in MBZ disposition, with a substantial decrease in AUC and Cmax. In addition, three polymorphs, A, B, and C, of MBZ exist, showing different solubility, toxicity, and therapeutic effects in anthelmintic applications [71,72,73]. Ren-Yuan Bai et al. [74] determined the pharmacokinetics of MBZ-A, B, and C and their concentrations in the brain and brain tumor distribution after oral MBZ administration on GL261 tumor-bearing mice. They suggest that MBZ-C is the most efficacious polymorph in brain tumor therapy with limited toxicity. Moreover, they demonstrated that the combination of Mebendazole-C with elacridar (P-glycoprotein inhibitor) could significantly improve the efficacy for future Phase I clinical trials of high-grade glioma and medulloblastoma. The high intra- and inter-patient variability may be essential to evaluate the response of MBZ as a potential anticancer therapy. Therefore, further research should elucidate the pharmacokinetic and dynamic profiles of upcoming formulations. To this end, monitoring the levels of MBZ and its metabolites should become standard practice in clinical treatments. Currently, preclinical evidence shows that chronic and high-dose schedules achieve plasma levels in the range required for clinical activity.

## 4. Preclinical Evidence of Mebendazole in Brain Cancer—In Vitro and In Vivo

Anthelmintic drugs have gained attention in the last two decades as potential anticancer agents due to their interactivity with microtubules [23,24,75,76]. In particular, a wide range of cancer cells and animal models showed the possible effect of MBZ in inhibiting tumor cell growth through its ability to inhibit tubulin polymerization, leading to a lethal effect in rapidly dividing cells. The potential effect of MBZ in inhibiting cancer cell growth has been described in thyroid [77], gastrointestinal [78], breast [79], prostate [80], pancreatic [81], ovarian [82], colorectal [83,84], melanoma [85], head and neck [86], leukemia [87], and bile duct cancer [88]. Moreover, MBZ can modulate various cancer-associated pathways, including MAPK14, MEK-ERK, C-MYC, USP5/c-Maf, TNIK, XIAP, ELK/SRF, NFKB, MYC/MAX, E2F/DP1, TGF/SMAD, AP1, and STAT1/2, dependent on the specific cancer model (Figure 3) [23,24]. In addition, MBZ is relatively non-toxic for normal cells, whereas it increases its specific sensitivity in cancer cells. Regarding brain cancer, several studies (Table 3) demonstrated the antitumor properties of MBZ. In 2011, routine animal studies unexpectedly showed that fenbendazole (a benzimidazole anthelmintic used to treat pinworm infection) inhibited brain tumor engraftment. Based on this hint, Ren-Yuan Bai et al. [89] proved that MBZ offers an auspicious opportunity for the clinical application of GBM; their research with in vitro and in vivo experiments with benzimidazoles identified for the first time MBZ as the most promising drug for GBM therapy. In detail, they observed that MBZ induced apoptosis in GBM cell lines. The IC_50_ was 0.24 μM in the GL261 mouse glioma line and 0.1 μM in the 060919 human GBM. In vitro activity was also correlated with significant inhibition of tubulin polymerization at 0.1 µM. They also proved that MBZ significantly extended mean survival to 65 days, compared with 48 days for the control group in syngeneic and xenograft orthotopic mouse glioma models. Moreover, they showed that the combination of MBZ plus temozolomide (TMZ) extends survival further than TMZ alone in the GL261 mouse model. Other diverse mechanisms of MBZ in CNS tumors have been proposed. In particular, Ren LW et al. [90] have suggested that all benzimidazole compounds (flubendazole, Mebendazole, fenbendazole) could inhibit the proliferation and metastasis of GBM cells regulating cell migration, cell cycle, programmed cell death, and other biological processes. Furthermore, MBZ inhibited the migration and invasion of GBM cells and regulated the expression of crucial (epithelial-to-mesenchymal transition) EMT markers, showing that MBZ could inhibit the metastasis of GBM. It also dose-dependently arrested the cell cycle at the G2/M phase of GBM cells through the P53/P21/cyclin B1 pathway.

The greatest challenge facing any CNS-targeted drug discovery program is the effective penetration of the blood–brain barrier (BBB). The limited ability of cancer therapeutics to accumulate in the tumor is the major obstacle to improving brain cancer therapy [91]. It is estimated that only ~2% of small-molecule drugs can effectively cross the BBB. MBZ’s small size (295 Daltons) and lipophilic property favor brain penetration. Depending on crystallization conditions, MBZ can form three different polymorphs, A, B, and C, which display significant differences in bioavailability. Ren-Yuan Bai et al. [74] studied MBZ’s brain penetration and pharmacokinetics and the therapeutic differences of the three polymorphic forms (A, B, C) on intracranial murine GL261 glioma allografts and human medulloblastoma D425 xenografts. The authors determined that MBZ-A showed low blood and brain concentrations and no antitumoral efficacy; however, MBZ-B and MBZ-C are therapeutically favorable for brain tumor therapy. Polymorph C demonstrated the highest penetration capability in the brain tissue and tumor. Furthermore, combination therapy of elacridar (P-glycoprotein (P-gp) inhibitor) and MBZ-C increased the survival in GL261 syngeneic glioma and D425 xenograft medulloblastoma models. In addition, De Witt M et al. [92] examined the mechanisms of tumor cell-killing of MBZ and cell viability compared with those of vincristine on GL261 glioblastoma cells. They demonstrated that MBZ and vincristine have similar inhibitory effects on cell viability and microtubule polymerization. They also compared the therapeutic efficacies and toxicities of MBZ and vincristine in the GL261-C57BL/6 syngeneic orthotopic mouse model, showing that MBZ is more effective than vincristine and provides significant survival.

Recently, Dakshanamurthy and colleagues [93] applied a computational proteo-chemometric method to a library of FDA-approved compounds and identified MBZ as a potential inhibitor of vascular endothelial growth factor receptor 2 (VEGFR2), which suggested a possible role of MBZ in interfering with tumor angiogenesis. VEGF is an essential pathway in angiogenesis-related glioma pathology; in glioblastoma, VEGF is upregulated and therefore induces angiogenesis with consequent production of dysfunctional and immature vessels, associated with significant edema and destruction of the BBB [94]. For this reason, therapeutic inhibition of VEGFR-2 action has a significant impact on restricting cancer growth. Numerous studies confirm that MBZ can suppress the autophosphorylation of VEGFR-2 kinase by competing with ATP. In this context, Ren-Yuan Bai et al. [95] tested the drug on a panel of eight medulloblastoma cell lines, obtaining IC_50_ for cell growth of 0.13–1 µM. They found that MBZ also inhibited vascular endothelial growth factor receptor 2 (VEGFR2) autophosphorylation at 1–10µM, while it cultured HUVECs with an IC_50_ of 4.3 µM. They demonstrated that MBZ selectively exhibited tumor angiogenesis but not normal brain vasculatures in orthotopic medulloblastoma models and interfered with VEGFR2 activity in multiple models. MBZ was able to prolong the median survival from 21 days to 48 days in the D425 xenograft medulloblastoma. Analysis of tumor sections from treated mice revealed a significant reduction of tumor angiogenesis, while the microvessel density in the normal brain parenchyma was unaffected.

A growing body of evidence suggests that activated Hedgehog (Hh) signaling is preclinically responsive to MBZ. Some studies indicate that the aberrant activation of Hh signaling leads to tumor cell growth, proliferation, and invasion. Therefore, the development of therapy targeting Hh signaling is an attractive and validated therapeutic strategy for treating a wide range of cancers [96]. Larsen et al. [97] found that MBZ is a potent inhibitor of the Hh signaling cascade in human medulloblastoma cultured cell lines and decreased GLI1 expression with an IC_50_ of 516 nM. Cell proliferation was inhibited at a concentration as low as 100 nM, and viability was significantly impaired at one µM. In the same paper, the authors suggested that MBZ treatment prevented the primary cilium formation. This microtubule-based organelle functions as a signaling hub for Hh pathway activation, decreasing the expression of downstream Hh pathway effectors and the proliferation and survival of human medulloblastoma cells with constitutive Hh activation. The activity of MBZ in vivo was then assessed in a DAOY intracranial mouse xenograft. Treatment significantly increased survival from 75 days in the control group to 94 days in the group administered MBZ 25 mg/kg and 113 days in the 50 mg/kg group. Along this same line, the study by Bodhinayake et al. [98] highlighted that using orthotopic models of medulloblastoma (a genetic model of the sonic hedgehog (SHH)allograft, SHH vismodegib-resistant, and D425 implanted into cerebellum) the MBZ treatment induced markedly extended survival of 150%, 100%, and 129%, respectively.

MBZ can also sensitize cancer cells to conventional therapy, such as chemotherapeutics and radiation, enhancing their combined antitumor potential, confirming that MBZ may be useful as an adjuvant therapeutic combined with traditional chemotherapy. Skibinski et al. [99] demonstrated that MBZ administration alone or combined with radiation effectively extended survival in preclinical models of malignant meningioma for the first time. The human meningioma cell lines treated with MBZ alone had IC_50_ values in the 0.26–0.42 μM range, similar to those observed in medulloblastoma and glioblastoma. The authors also described a synergistic effect when combining MBZ therapy with radiation in an intracranial mouse model of malignant meningioma. This combination increased median survival and delayed tumor growth by inducing apoptosis via the caspase-3/7 pathway, decreasing cell proliferation and reducing levels of the angiogenesis marker CD31. Another study observed significant improvement in the radiosensitization of glioma cells (GL261 and GBM14) after MBZ treatment. Markowitz et al. [100] treated GL261 and GBM14 glioma cells with MBZ 3–9 h post-IR; this experiment demonstrated that MBZ could sensitize cancer cells to IR independently of the induction of mitotic arrest.

Several studies have shown that MBZ and TMZ coadministration led to a synergic response by glioblastoma cells, with an increase in mortality both in sensitive and resistant cells to TMZ. Glioblastoma patients, whose tumors expressed low FGFR3 and AKT2, responded poorly to TMZ. Kipper et al. [101] suggested that the triple combination of Temozolomide, Vinblastine, and Mebendazole (TVM) may be a considerable therapeutic alternative for the TMZ-tolerant gliomas that harbor low expression of FGFR3/AKT2. All cell lines showed an increase in Notch3 expression after TVM treatment.

In a recent study, Ariey-Bonnet et al. [102] confirmed that MBZ decreased the viability of glioblastoma cells in vitro. Moreover, for the first time, they found that the cytotoxic activity of the MBZ is significantly correlated with its ability to inhibit MAPK14 kinase activity in vitro, which relates to poor prognosis in GBM patients, increased tumor invasiveness, and aggressive phenotype [103,104]. This study suggests that targeting MAPK14 with MBZ is a promising strategy to enhance chemotherapy efficacy against GBM cancer.

## 5. Clinical Evidence of Antitumoral Properties of Mebendazole

Clinical trials can help clarify the availability of MBZ as adjuvant and neoadjuvant therapeutics for cancer treatments. Six complete or recruiting clinical trials investigating the anticancer effect of MBZ, alone or in combination with other drugs, are currently registered at clinicaltrials.gov and are listed in Table 4.

In 2011, Dobrosotskaya et al. [105] published the first clinical case report on MBZ as a cancer treatment in a human patient. He described a 35-year-old woman affected by metastatic adrenocortical carcinoma (right adrenal gland and multiple liver metastases) who showed tumor progression after repeated surgeries, radiation, and chemotherapy treatments. After administering MBZ 100 mg orally twice daily for 19 months, they observed prolonged tumor response, as the liver metastases initially regressed and remained stable for 19 months. In contrast to the morbidity observed with other treatments, MBZ was well tolerated and greatly improved the patient’s quality of life. This study also demonstrated reduced angiogenesis in the tumor treated with MBZ, which may result from diminished tumor growth. However, 24 months after the beginning of oral MBZ, a scan showed disease progression, and everolimus was added to the MBZ but without additional benefit in disease control.

Nyger and Larsson [106] reported that MBZ induced remission of lung and lymph node metastases and a partial remission of liver metastases in a patient with refractory colon cancer. The authors considered the case of a 74-year-old patient affected by metastatic colon cancer in progression at multiple sites (lungs, abdominal lymph nodes, and liver), following two lines of chemotherapy (with capecitabine, oxaliplatin, and bevacizumab, and then capecitabine and irinotecan). They had no other standard treatment options, so they started a treatment based on an oral dose of MBZ of 100 mg twice daily. After six weeks of monotherapy of MBZ treatment, the patient showed near-complete remission of the metastases in the lungs and lymph nodes and a good partial remission in the liver after the computerized tomography evaluation. The liver enzymes AST (serum aspartate aminotransferase) and ALT were found to be elevated from five to seven times, so MBZ was temporarily stopped and reintroduced at half dose. Liver enzymes slowly decreased, and the patient reported no adverse effects from MBZ; moreover, the disease remained stable. After ceasing treatment for about three months, the patient developed brain metastases treated with radiotherapy, followed by evidence of disease in the lymph nodes.

A phase I clinical study started in 2013 and was completed in 2021 at John Hopkins Hospital; the goal was to determine the maximum tolerated dose (MTD) of MBZ in combination with TMZ after surgery and the standard radiation and TMZ treatment in patients with newly diagnosed malignant gliomas. The outcome supports that oral MBZ can be used safely in high doses in combination with TMZ (http://clinicaltrials.gov/ct2/show/NCT01729260, accessed on 24 October 2022). In this study, 24 patients (18 glioblastomas and six anaplastic gliomas) received MBZ in combination with adjuvant TMZ after completing concurrent radiation plus TMZ. The primary aim was to determine the maximum tolerated dose of MBZ with TMZ and whether this combined regimen could slow tumor progression. Dose-escalation levels were 25, 50, 100, and 200 mg/kg/day of oral Mebendazole. A total of 15 patients were enrolled at the highest dose studied with 200 mg/kg/day. This dosing is based on a previous study of 37 children treated for hydatid disease with 100–200 mg/kg/day of MBZ without serious side effects [51]. Although the 200 mg/kg/day dose level encountered late toxicity, it is not considered MTD, as none of the AEs (those requiring hospitalization or resulting in death) were attributable to MBZ. However, there are considerations as to why levels less than 200 mg/kg/day might be preferable. The most common side effect was the reversible elevation of liver enzymes. In conclusion, the authors affirm that further clinical evaluation of MBZ in patients with high-grade gliomas is warranted to evaluate better this regimen’s potential benefit.

Currently, a clinical trial phase I/II investigates the anticancer effects of MBZ in combination with standard-of-care agents for recurrent pediatric brain cancers that are not responding to standard therapies (http://clinicaltrials.gov/ct2/show/ NCT02644291). The patients for this experimental trial were aged 1 to 21 years with a diagnosis of medulloblastoma, or high-grade glioma (pediatric glioblastoma, anaplastic astrocytoma, and diffuse intrinsic pontine glioma), where the tumor has resumed growth or continued to grow despite standard medical therapy. The study patients will be divided into two groups (low-grade glioma and high-grade/pontine glioma) to determine the maximally tolerated dose of MBZ.

Patients on the low-grade arm will receive treatment with seven 10-week cycles of carboplatin, vincristine, temozolomide, and MBZ. In comparison, patients on the high-grade glioma/pontine glioma arm will receive treatment with twelve 28-day cycles of bevacizumab, irinotecan, and MBZ. In addition, to determine the safety and efficacy of these combinations, MBZ (50 mg/kg/day, 100 mg/kg/day, or 200 mg/kg/day) will be given orally twice daily throughout treatment (70 weeks for low-grade glioma patients, 48 weeks for high-grade glioma/pontine glioma patients).

The second clinical trial is a phase I and II pilot study of MBZ in combination with vincristine, carboplatin, and temozolomide for pediatric patients with low-grade gliomas at Cohen Children’s Medical Centre of New York (http://clinicaltrials.gov/ct2/show/NCT01837862). Patients with low-grade gliomas will receive a regimen of MBZ in combination with vincristine, carboplatin, and temozolomide. In contrast, patients with high-grade gliomas and diffuse intrinsic pontine gliomas will receive a regimen of MBZ in combination with bevacizumab and irinotecan. MBZ doses will be escalated from the initial dose level of 50 mg/kg/day to a second dose of 100 mg/kg/day and a third of 200 mg/kg/day, each divided twice daily. A standard “3 + 3” design will be used for determining dose escalation. The primary objective of the Phase I part of the trial is to determine if MBZ is tolerated when used in combination with the current three-drug regimen. After determining the maximally tolerated dose for each group, the study will continue to evaluate the efficacy of this regimen. The study will be corrected for the maximally tolerated dose for each group; it will be used in the remainder of the study. Phase II aims to include the time of progression-free status in patients and their overall survival. Currently, the study is recruiting participants, and the estimated study completion date is April 2025.

## 6. Conclusions

The repurposing of drugs not initially developed for cancer treatment is an ongoing research activity in many countries worldwide. The recognition that drug repositioning is an excellent opportunity to expedite anticancer therapies to the clinic in oncology represents a solution to the exorbitant cost of discovering new drugs and has the substantial advantages of cheaper, faster, and safer preclinical and clinical validation protocols. The consolidated knowledge of preclinical and clinical data on pharmacokinetics, side effects, and regimens may accelerate approval. Moreover, repurposing such compounds in oncology may also elude the detrimental impacts of conventional cancer chemotherapeutics, which may produce adverse effects that negatively affect patients’ quality of life [107,108,109,110,111].

MBZ is attractive as an anticancer therapy because it can cross the blood–brain barrier permeability with a low-toxicity profile. Toxicity is particularly low in children compared to other microtubule inhibitors such as vincristine and paclitaxel. Despite its poor oral bioavailability, available preclinical studies demonstrated that MBZ reaches plasmatic and tissue concentrations sufficient to activate antineoplastic activity in vitro. Additional clinical investigations will clarify this issue.

Many research groups have investigated MBZ as a tubulin polymerization inhibitor and strongly recommend its clinical use as a replacement for vincristine for treating brain tumors [92]. In addition, many studies demonstrated its capability in reducing angiogenesis, arresting the cell cycle, and targeting several key oncogenic signal transduction pathways. Therefore, MBZ’s ability to hit multiple targets can improve the efficacy of anticancer therapy and help overcome acquired resistance to conventional chemotherapy.

One of the most relevant MBZ properties is that it exerts cancer cell-specific selectivity inducing minimal cytotoxicity in normal cells while inducing high cytotoxicity in tumor cells, showing a favorable therapeutic index for in vivo applications [89,100,101]. Finally, MBZ can inhibit endothelial cells and tumor angiogenesis by inhibiting VEGF receptor 2. Many studies demonstrated the anti-angiogenic effect of MBZ in medulloblastoma preclinical mouse models and its encouraging impact on overall survival [95,98].

Although MBZ has several advantages, researchers must address some challenges before its successful use for clinical purposes. Potential limitations for successfully repositioning MBZ in oncology relate to its physicochemical properties, route of administration, and poor bioavailability with significant individual pharmacokinetic variability [51,55,70].

Furthermore, achieving sufficient efficacy of repurposed MBZ for cancer therapy may require treatment at higher doses and for more extended periods than the conventional prescription, resulting in unexpected side effects. Researchers are investigating diverse formulation strategies and fatty meals to improve oral absorption and reach therapeutic blood levels to increase drug solubility and dissolution rate. For this reason, further studies should focus on the poor aqueous solubility and systemic availability of MBZ because its improvement may enhance anti-tumorigenesis and prolong overall survival.

Although MBZ has already been under phase II/III clinical trials for cancer therapy, there are still too few experimental studies in the clinical setting for translating the preclinical study knowledge to the patient context. However, the few available experimental studies confirm the sufficient safety and efficacy of MBZ in tumor patients. MBZ has prolonged patient survival and improved patient outcomes in some clinical trials, supporting its application’s feasibility in a clinical setting. A few case reports demonstrated that patients with metastatic late-stage cancers responded to MBZ, showing reduced metastasis and stabilized disease. Ongoing clinical trials will clarify the potential use of MBZ in brain cancer treatments.

In summary, the available data indicate that the repositioning of MBZ in treating brain cancer is promising and may represent a low-cost and fast route for expanding approved drugs for better brain cancer therapy. In the future, MBZ could be an alternative drug for treating TMZ-sensitive GBM patients alone or in combination with TMZ. Further studies are needed to confirm the potential use of MBZ in patients with TMZ-resistant GBM.

## Figures and Tables

**Figure 1 ijms-24-01334-f001:**
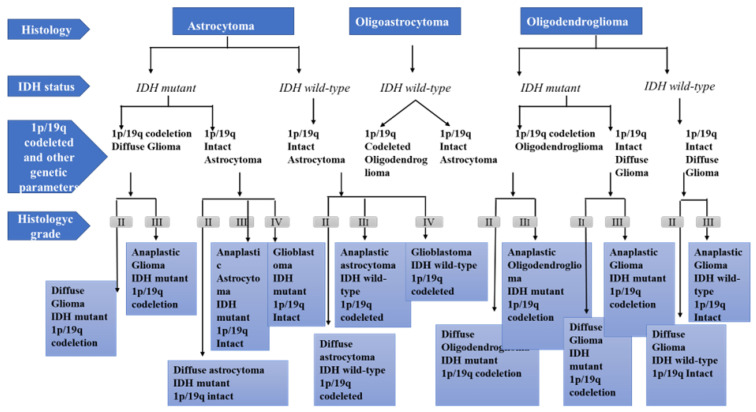
Diagnostic algorithm for diffuse gliomas, based on the WHO classification.

**Figure 2 ijms-24-01334-f002:**
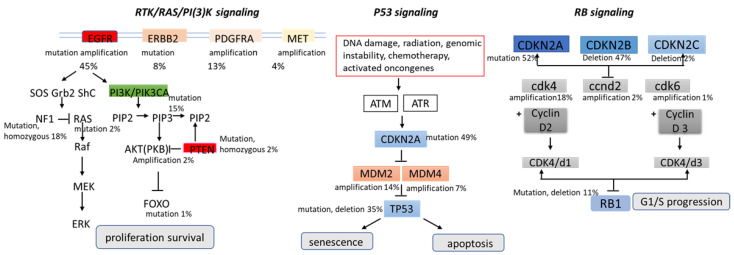
Major signaling pathways involved in the pathogenesis of glioblastomas.

**Figure 3 ijms-24-01334-f003:**
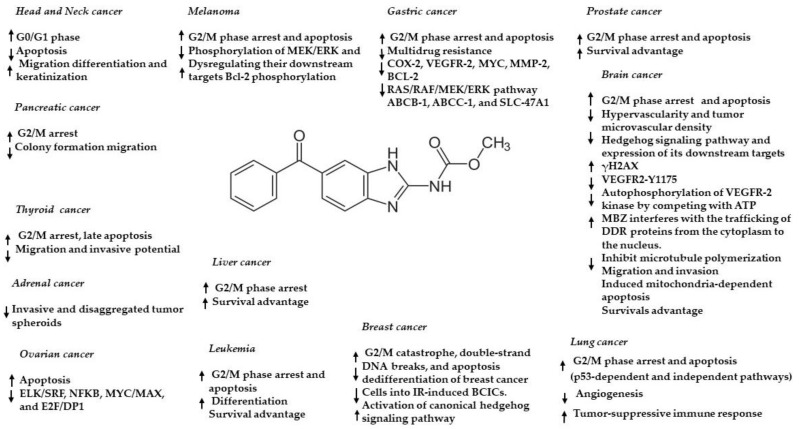
Anticancer effects and mechanisms of action of MBZ in different cancers. ↑ upregulation/activation; ↓ downregulation/inhibition.

**Table 1 ijms-24-01334-t001:** Chemical properties of MBZ.

Mebendazole:	Methyl 5-benzoyl-1H-benzimidazol-2-yl-carbamate
Structure	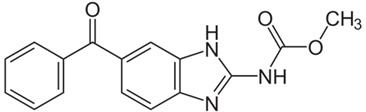
Chemical formula	C16-H13-N3-O3
Weight	295.293 g/mol
CAS	60254-95-7
Indication	oral administration
Absorption	poor solubility and absorption
Distribution	highly bound to plasma protein
Metabolism	extensively hepatic first-pass
Side effects	gastrointestinal upset, diarrhea, fever, abdominal
	discomfort, flatulence, hypersensitivity reactions
Excretion	bile and feces and ˂2% urine
BBB permeability	Yes
GI Absorption	High
P-gp Substrate	No
T_1/2_: elimination half-life	3–6 h

**Table 2 ijms-24-01334-t002:** Mean pharmacokinetic parameters of MBZ extracted from different studies.

Author (Year)	Population	Formulation	C_max_ (ng/mL)		AUC (ng.h/mL)		T_max_ (h)		T_1/2_ (h)		MBZ Dose and Coadministration with Other Drugs	Ref
Brainthwaite (1982)	Patients (n.10)	tablet	17.2–116.2		209.6		5.2				MBZ 10 mg/kg	[55]
	Chronic treatment		99.4–500.2		649.8		2				MBZ 10 mg/kg	
Dawson (1985) Edwards (1988)	Fasting volunteers (n.3)	oral intravenous	91, 1–142		4.63 ± 1.80 2.27 ± 8.2		0.42 ± 0.12		0.93 ± 0.25 1.12 ± 0.24		MBZ 1.18 µg of ^3^H-mebendazole	[47]
Corti (2009)	Healthy volunteers (n.8) Fasting	tablet	31.0 ± 26.0		207.2 ± 157.6		2.1 ± 1		7.4 ± 2.2		MBZ 1000 mg/day po on 3 days	[56]
		tablet	36.0 ± 22.8		228.9 ± 147		2.4 ± 1.7		9.3 ± 3.5		MBZ 1000 mg/day po on 3 days + Short-term Ritovaris 200 mg po twice daily × 2 doses	
		tablet	11.5 ± 6.2		85.9 ± 53.2		2.1 ± 0.8		10.6 ± 8.6		MBZ 1000 mg/day po on 3 days + Long-term Ritovaris 200 mg po twice daily × 7 doses	
Bekhti (1987)	Patients (n.8) with peptic ulcer and hydatid cysts	tablet	55.7 ± 30.2								MBZ 1.5 g three times daily po × 30 days	[57]
			82.3 ± 41.8								MBZ 1.5 g three times daily po × 30 days + Cimetidine 400 mg three times daily po × 30 days	
Luder (1986)	Patients (n.10) with Echinococcus multilocularis or granulosus	tablet	HC								MBZ 45–122 mg/kg/day po × 10 weeks + Cimetidine 1 G/day po × 10 weeks	[58]
			NC								MBZ 45–122 mg/kg/day po × 10 weeks + Ursodeosycholic acid 500 mg/day	
Liu (2012)	Mouse	powder	1.3 ± 0.42		11.6 ± 2.0		1.3 ± 0.6		12.0 ± 5.5		MBZ-1% tragacanth po 25 mg/kg	[59]
			3.3 ± 0.6		28.2 ± 2.5		1.7 ± 1.6		11.5 ± 6.2		MBZ-Oleic Acid po 25 mg/kg	
			4.8 ± 0.6		19.8 ± 2.5		1		4.2 ± 1.0		MBZ-Glycerol Trioleate po 25 mg/kg	
			4.4 ± 2.0		25.1 ± 4.4		1.5 ± 0.9		4.9 ± 1.7		MBZ-Soybean oil po 25 mg/kg	
Ren-Yuan Bai (2015)	MBZ polymorphs in mice (GL261 Glioma model)	tablet	Plasma	Brain	Plasma	Brain	Plasma	Brain	Plasma	Brain		[53]
			379.3		3052		1		3.23		MBZ Polimorph A	
			2778.3		26474		6		3.18		MBZ Polimorph B	
			2553.3	2016	16039	13134	4	4	0.90	1.64	MBZ Polimorph C	

AUC: area under the concentration-time curve; Cmax: maximum drug concentration; T1/2: elimination half-life; Tmax: time to reach Cmax; HC: higher concentration; NC: no change; N: sample size.

**Table 3 ijms-24-01334-t003:** The most relevant molecular properties of MBZ in preclinical studies.

Cancer Type	Cell Lines	Model Used	Molecular Targets	Efficacy	IC50 or Doses	Combination with Other Drugs	Ref
Glioblastom	060919	In vitro	Inhibit tubulin polymerization	↓ Cell viability	0.11 μM		[57]
		In vivo orthotopic mouse models	Apoptosis	↑ Survival: 48 d CT: 65 d MBZ			
Glioma	GL261	In vitro		↓ Cell viability	0.24 µM		
		In vivo orthotopic mouse models		↑ Survival: 30 d CT: 49 d MBZ ↑ Survival: 29 d CT: 41 d TMZ: 50 d MBZ + TMZ		Temozolomide	
Glioma	GL261	In vivo orthotopic mouse models	Inhibit tubulin polymerization	↑ Survival (MBZ tablets from different suppliers) 29 days CT; 34 days in S2015; 50 days in S2017; 42 days in medley; 44 days in Janssen	50 mg/kg		[53]
Medulloblastoma	GL261 D425	In vivo orthotopic mouse models		↑ Survival with elacridar		Elacridar (P-glycoprotein inhibitor)	
Glioma	GL261	In vitro	MBZ sensitizes GL261 cells to IR	↓ Cell viability	35 nM	IR (ionizing radiation)	[79]
Glioblastoma	GBM14 glioma cells		MBZ sensitizes GBM14 cells to IR	↓ Cell viability		IR (ionizing radiation)	
Glioblastoma	U87-MG	In vitro	Inhibited migration and invasion	↓ Cell viability	0.21 μmol/L		[58]
	U251-MG		Arrest the cell cycle at the G2/M phase	↓ Cell viability	0.25 μmol/L		
Glioblastoma	GL261	In vitro	Tubulin disruption	↓ Cell viability	160 nM		[71]
		In vivo orthotopic mouse models		↓ Tumor growth ↑ Survival: CT 10 d; 50 MBZ 14 d; 100 MBZ 16.5 d No effect with vincristine	50 mg/kg 100 mg/kg		
Medulloblastoma	HUVEC	In vitro	↓VEGFR2 kinase activity, CD31		1–10 µM 4.3 µM		[74]
	D425	In vivo orthotopic mouse models		↓ Tumor growth ↑ Survival: CT 21 d; MBZ 48 d	50 mg/kg		
		PTCH Mutant allograft MB		↑ Survival: CT 12 d; MBZ 30 d	50 mg/kg		
		PTCH Mutant D477G allograft MB		↑ Survival: CT 12 d; MBZ 30 d	50 mg/kg		
Medulloblastoma	Daoy	In vitro	Inhibits hedgehog Signaling pathway ↓ GLI1	↓ Cell viability and colony formation	516 nmol/L		[76]
	Daoy	In vivo orthotopic mouse models	↓ GLI1 and PTCH2	↓ Tumor growth and hedgehog signaling ↑ Survival: CT 75 d; MBZ 113 d	50 mg/kg		
Medulloblastoma	D425 SHH SHH- vismodegib-resistant	In vivo orthotopic mouse models	inhibition of VEGFR2 kinase activity	↑ Survival: CT 21 d; MBZ 48 d	50 mg/kg		[77]
Meningioma	KT21MG1 IOMMLE AC-1 SF4068 SF6717, SF1335 SF1335 + YAP	In vitro	Apoptosis induction, angiogenesis Inhibitor	↓ Cell viability and colony formation	0.39 μM 0.39 μM 0.342 μM 0.42 μM 0.372 μM 0.262 μM		[78]
	KT21MG1	In vivo orthotopic mouse models	↓ Tumor, Ki67, CD31 ↑ Survival cleaved caspase 3	↑ Survival: CT 19 d; MBZ 30 d; radiation 33.5 D; MBZ + R 39 d	50 mg/kg	Radiation	
Glioblastoma	U87 A172 U251 U138 C6 (murine)	In vitro	Low expression of FGFR3 and AKT2 was especially sensitive to T + V + M.	↓ Cell viability	500 nM	50 µM TMZ 5 nM VBL	[80]
				↓↓↓ Cell viability	T + V + M		
Glioblastoma	U87 U87vIII T98G U251	In vitro	MBZ inhibits ABL1, ERK2/MAPK1, and MAPK14/p38a in vitro, with particularly high potency against MAPK14/p38a	↓↓↓Cell viability	2.1 μM 288 nM		[81]

**Table 4 ijms-24-01334-t004:** Clinical trials of MBZ in cancer patients.

Title	Phase	Conditions	Intervention/ Treatment	Institution	Study/Results	Status
Study of Mebendazole in Newly Diagnosed High-Grade Glioma Patients Receiving Temozolomide NCT N. NCT01729260	Phase 1	Newly diagnosed high-grade glioma (WHO Grade III or IV)	Mebendazole: 500 mg chewable tablets with meals three times every day on a 28-day cycle	The Johns Hopkins Hospital Baltimore, Maryland, United States, 21287	No results Available	Study Start Date: 2012 Study Completion Date: 2021 Completed
Phase I Study of Mebendazole Therapy for Recurrent/Progressive Pediatric Brain Tumors NCT N. NCT02644291	Phase 1	Medulloblastoma Astrocytoma, Grade III Glioblastoma Anaplastic Astrocytoma Brain Stem Neoplasms, Malignant Oligodendroblastoma Anaplastic Oligodendroglioma Malignant Glioma	Mebendazole: 500 mg tablets, three divided doses with meals	Johns Hopkins All Children’s Hospital, Saint Petersburg, Florida, United States, 33701 Johns Hopkins University School of Medicine Baltimore, Maryland, United States, 21231	No results Available	Study Start Date: 2016 Study Completion Date: June 2022 Completed
A Phase I Study of Mebendazole for the Treatment of Pediatric Gliomas Phase I: determine if MBZ is tolerated when used in combination with the current three-drug regimen. Phase II: evaluate the efficacy of this regimen. NCT N. NCT01837862	Phase 1 Phase 2	Pilomyxoid Astrocytoma Pilocytic Astrocytoma Glioma, Astrocytic Optic Nerve Glioma Pleomorphic Xanthoastrocytoma Glioblastoma Multforme Anaplastic Astrocytoma Gliosarcoma Diffuse Intrinsic Pontine Glioma (DIPG) Low-grade Glioma Brainstem Glioma	Mebendazole: 50 mg/kg/day, 100 mg/kg/day, or 200 mg/kg/day p.o. and b.d. for 70 weeks for Low-grade Glioma (in combination with vincristine, carboplatin, and temozolomide) and 48 weeks for High-grade Glioma/Pontine Glioma (in combination with bevacizumab and irinotecan)	Cohen Children’s Medical Center of New York Recruiting New Hyde Park, New York, United States, 11040	No results Available	Study Start Date: 2013 Estimated Primary Completion Date: April 2024 Estimated Study Completion Date: April 2025 Recruiting
A Clinical Safety and Efficacy Study of Mebendazole on GI Cancer or Cancer of Unknown Origin NCT N. NCT03628079	Phase 1 Phase 2	Cancer of the Gastrointestinal Tract Cancer of Unknown Origin	ReposMBZ Capsules 50 mg, 100 mg, 200 mg Pharmacokinetics analysis	Dept of Oncology, University Hospital, Uppsala, Sweden, 75185	No Results Available	Study Start Date: 2018 Study Completion Date: 2019 Terminated
Clinical Study Evaluating Mebendazole as Adjuvant Therapy in Patients with Colorectal Cancer NCT N. NCT03925662	Phase 3	Colorectal Cancer	Folfox with Avastin only Folfox with Avastin with Mebendazole	Sherief Abd-Elsalam, Tanta University	No Results Available	Study Start Date 2019 Study Completion Date: 2028 Recruiting
Study of the Safety, Tolerability, and Efficacy of Metabolic Combination Treatments on Cancer (METRICS) NCT N. NCT02201381	Phase 3	Cancer Overall Survival	Oral Mebendazole 100 mg p.o. and uid, for the study duration Oral atorvastatin up to 80 mg uid Oral metformin up to 1000 mg uid Oral doxycycline 100 mg uid	Care Oncology Clinic London, United Kingdom, W1G 9PP		Study Start Date: May 2022 Study Completion Date: 2027 Withdrawn (Prospective recruitment not possible)
Mebendazole Monotherapy and Long-Term Disease Control I Metastatic Adrenocortical carcinoma. 2011	Clinical Study	48-year-old man with adrenocortical carcinoma	Mebendazole: 100 mg twice daily for 19 months	University of Michigan	Metastases regressed, and the disease remained stable Well tolerated, and the associated adverse effects are minor	
Drug repositioning from bench to bedside: Tumor remission by the anthelmintic drug Mebendazole in refractory metastatic colon cancer. 2013	Clinical Study	74-year-old man with metastatic colon cancer	Repositioning drugs for use in advanced colon cancer Mebendazole: 100 mg twice a day b.d. for six weeks	Uppsala University, Sweden	Complete remission of the metastases in the lungs and lymph nodes and a good partial remission in the liver No adverse effects from the treatment	

## Data Availability

Not applicable.

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
