# Peer review of "Emerging Perspectives on the Antiparasitic Mebendazole as a Repurposed Drug for the Treatment of Brain Cancers"

_ijms, 2023, doi:10.3390/ijms24021334_

Round 1
Reviewer 1 Report (New Reviewer)
The authors wrote a review focused on the anticancer effects of mebendazole (MBZ), a small molecule drug which is primarily used in the clinic as a broad-spectrum anti-helminthic drug. In addition to its well-established anti-helminthic activity, MBZ has also been studied as a potentially cheap alternative for oncologic indications based on its in vitro activity against a variety of cancer cell lines and in vivo efficacy in a number of animal tumor models, which were at least partly validated by some clinical data. The cheap cost of manufacturing and its excellent safety profile recommends mebendazole as a viable alternative to current chemotherapeutics or targeted small molecule drugs. The primary mechanism of action for MBZ is the targeting of microtubular architecture in parasites and mammalian cells by blocking the polymerization of β-tubulin into microtubules, which further leads to the disruption of the vesicular system and the inhibition of glucose uptake and transport. Moreover, MBZ has been shown to modulate several signaling pathways that are known to operate aberrantly in the microenvironment of multiple types of cancer including gliomas. All these activities have important consequences on tumor cell proliferation and motility and on tumor microenvironmental processes such as angiogenesis.
However, there are still a number of challenges that temper the enthusiasm when it comes to the development of mebendazole for oncologic indications. The biggest issue with the drug remains the poor oral bioavailability due to its abysmal absorption across the intestinal mucosa, which makes the drug an excellent solution when it comes to targeting luminal parasites but a poor one for reaching tumors in distant organs. The obvious solution to this is the development of drug derivatives with improved bioavailability, but this would mean the development of entirely new drug products which are expected to come at steep additional costs. While the lipophilicity of MBZ is a plus when it comes to the ability of the drug to cross the blood-brain barrier, currently no robust data is available to demonstrate effective therapeutic concentrations for the drug in the brain tumor microenvironment (or at least in the cerebrospinal fluid) after oral administration. When it comes to tumors outside the CNS, two very exciting case reports have been published in the past with MBZ which suggest that the drug has the ability to regress metastatic disseminations systemically. On a more cautionary note, clinical trial data available for brain oncologic indications is still very limited with only a few ongoing clinical trials currently in brain cancer and virtually no results released publicly from already completed trials.
The present review is generally well written, and the authors provide a good summary of the current development of mebendazole for clinical indications such as brain cancers. The authors reviewed some of the exciting aspects of repositioning mebendazole as an oncologic drug and also discussed some of the drawbacks and challenges faced by the drug. However, there are a few instances where I would have preferred to see a more cautionary tone regarding certain assertions. For instance, lines 553-554 read “Moreover, despite its poor oral bioavailability, MBZ reaches plasmatic and tissue concentrations sufficient to activate antineoplastic activity in vitro”. I am not entirely sure if the current clinical data supports this assertion for brain tumors. Also, lines 596-597 read “Patients with metastatic late-stage cancers responded well to MBZ, showing reduced metastasis and stabilized disease”. To my knowledge, there are only two case reports supporting this assertion and no data is currently available from already completed clinical trials. Lastly, lines 605-606 read “In the future, the MBZ could be utilized as an alternative drug for treating TMZ-resistant GBM patients”. To my knowledge, most in vitro and in vivo animal model data with MBZ was generated with GBM cell lines that are TMZ sensitive while very scant data is currently available from TMZ-resistant models. Therefore, I am not entirely sure if the last assertion made by authors regarding TMZ-resistant GBM is properly supported by the current evidence.
Author Response
Please, see the attachment

Reviewer 2 Report (New Reviewer)
Please edit the English. Some points that I have come across:
LINE 18 acronym: Membendazole (MBZ)
LINE 129 which were IDH status….. Should change to: according to the IDH status
Line 129 more precisely : The co-deletion of chromosome arms 1p and 19q or chromosome p/19q co-deletion.
Fig1 and 3 have low resolution
Line 222 and 460 axplain ALT, AST
Line 239-241 must rephrase
Line 242 must rephrase
Lin2 247 improve
Lines 262-265 must rephrase \line 283: as well known In addition.
line 442 rephrase
Author Response
Please, see the attachment

Reviewer 3 Report (New Reviewer)
Comment:
1) Importance & Novelty: The manuscript's purpose is to discuss recent developments in the Repurpose of Mebendazole (MBZ) for the Treatment of Brain Cancers, and its potential as a cancer treatment target is a topic of high importance and interest to a general audience. Given that the MBZ in the context of brain cancers and MBZ-orchestrated therapies have been previously reviewed, the present manuscript must stay focused on MBZ-targeted treatments for brain cancers, as stated in the title and the abstract. Indeed, the current manuscript focuses on discussing MBZ-targeted therapies for brain cancers and, as a result, is novel.
2) Argumentation & critical analysis: At times, the manuscript does a decent job at critically analyzing prior work, for example, discussing the limitations of MBZ-targeted therapies, noting whether the referenced studies were performed on in vitro or in vivo model systems or on human clinical samples, reporting the statistical significance of the referenced data, etc. However, the current version of the manuscript is inconsistent with such critical analysis of prior work and needs improvement in this regard. Furthermore, multiple arguments in the present manuscript are poorly supported or contradictory; references are sometimes missing.
3) Writing quality: this reviewer could follow the current version of the manuscript in skimming. However, upon attending to specific arguments, there were multiple poorly constructed and confusing clauses, inadequate background discussion, the sufficient transition.
4) Figures & Tables: The manuscript provided 4 tables with a short list of treatment schemes and clinical trials. The tables were neither exhaustive nor provocative to the present manuscript. The authors should include a figure that depicts MBZ in the context of brain cancer in illustrating, for example, tumor cell infiltrating, immune cell infiltration, target-mediated, etc. Notably, the figures should be focused on the specifics of brain cancers. Furthermore, the authors should include a figure that depicts various brain cancer-related pathways and their relationship of MBZ with other drugs; such a figure would greatly help the reader to follow the text.
5) In conclusion: The current version of the manuscript reads like many facts (data dumping style) piled up (regurgitated) from various papers like a laundry list of publications. They should flush out a specific problem derived from such thorough digestion of scientific facts, thereby pointing out that the predicament seems insurmountable and offering the solution of their insight and perspective to the situation in the manuscript. To tighten the grip over the logic flow, the authors should not crawl freely, crossing the boundaries of different concepts. A good literature review that is easy to follow has a solid frame. Unstructured, verbose, choppy writing can kill a legitimately good insight. The reader wants to spend the minimal amount possible to gain the information. So, the facts must be decomposed into well-organized, logical arguments supported by the selected evidence.
Thus, the current version is preliminary. After careful consideration, this reviewer felt that it has merit but is not suitable for publication as it currently stands. We uphold the standard and integrity of [IJMS] with rigorously constructive critiques, which indicated in an impact factor of 6.208, substantially higher than the average Journal of IF 1.0 out of 29,000 Journals collected by The Web of Science™ platform (Clarivate) that is the world's most trusted publisher-independent global citation database. Below are specific comments and suggestions for the authors that should be incorporated to improve its clarity, coherence, and logic flow.
Specific comments:
1) The current version of the title did not capture the content of the text: Neither insightful nor provocative to attract readers.
2) The abstract: Lines 26-29 should be expanded to elaborate on the authors' own insightful opinions into the field by cutting down the introduction of MBZ, which is well known – none of those lines convey much on why the readers are interested in reading on the rest. E.g., Lines 97 -98, "proliferation inhibition, cell cycle arrest, apoptosis induction, angiogenesis inhibition, and targeting several key oncogenic signal transduction pathways," should be uplifted onto the abstract.
3) Line 429: "Figure 3. Anticancer effects and mode of action of mebendazole in different types of cancer" should be integrated with Table 3, focusing on brain tumors, and Table 4 (three clinical trials). E.g., The mutual interplay of gut microbiota and brain tumors should be explored to offer new perspectives to MBZ functions [Science. 2020 May 29;368(6494):973-980. doi: 10.1126/science.aay9189. PMID: 32467386].
4) Lines 234-291: How did the authors conclude the in vivo pharmacokinetic and bioavailability in organ-specific and temporal distributions?
5) Fig 3, Lines 428-430: the authors entitled on brain tumors, so they need to expand on brain tumors in the figure.
6) How did the routes of administration (local vs. systemic) affect pharmacokinetic and bioavailability in organ-specific and temporal distributions? In particular, for Bacteria detected inside human tumors.
7) Table 4: NCT N. NCT01837862, what was the rationale for the combinations of multiple drugs with what dosing strategies?
8) Lines 465-466: "They did not recommence MBZ treatment after discovering brain metastases." What was the reason for the study, or what did the authors speculate?
9) Lines 536-608 were written instead as a conclusion, which should be a simple take-home message, but as a discussion. E.g. Lines 550-560 were written like an introduction.
10) Lines 561-574 were read like repetitive materials with citations, which didn't belong in the conclusion.
11) Line 578: what was the literature support of "poor bioavailability with significant individual pharmaco-kinetics variability" – did they follow any individual cases? How did any individual patients manifest in efficacy? Data? Mechanisms?
12) Lines 599 – 608 overstated the data sets.
13) Line 544: "may also elude the detrimental impacts of conventional cancer chemotherapeutics," was not fully explained and supported by citations.

Author Response
Please, see the attachment

Round 2
Reviewer 3 Report (New Reviewer)
Accepted as R2 fully addressed my comments.
This manuscript is a resubmission of an earlier submission. The following is a list of the peer review reports and author responses from that submission.
Round 1
Reviewer 1 Report
The authors report a review of the literature on the use of Mebendazole on brain tumors. Although it is a good idea to provide the readers of the journal with a review on this type of therapeutic alternative, the choice to generalize to "brain tumors" without clarifying the different mechanisms on the various pathogenesis (some of which are very different, such as between Glioblastoma and medulloblastoma) does not seem good enough for publication. The abstract appears to be generic and does not clarify the nature of the article. The introduction is not comprehensive and the discussion appears to be a copy-paste of the main clinical trials published in the literature. the summary table appears unclear. In my opinion, the paper does not appear suitable for publication.
Reviewer 2 Report
The review is quite interesting, but it would be more qualitive and attractive if the authors would pay more attention to some aspects of brain nerve tissue histology and GBM pathomorphology
